# Cas9+ conditionally-immortalized macrophages as a tool for bacterial pathogenesis and beyond

**Allison W Roberts[†], Lauren M Popov[†], Gabriel Mitchell, Krystal L Ching, Daniel J Licht, Guillaume Golovkine, Gregory M Barton, Jeffery S Cox\***

Department of Molecular and Cell Biology, University of California, Berkeley, Berkeley, United States

**Abstract** Macrophages play critical roles in immunity, development, tissue repair, and cancer, but studies of their function have been hampered by poorly-differentiated tumor cell lines and genetically-intractable primary cells. Here we report a facile system for genome editing in non-transformed macrophages by differentiating ER-Hoxb8 myeloid progenitors from Cas9-expressing transgenic mice. These conditionally immortalized macrophages (CIMs) retain characteristics of primary macrophages derived from the bone marrow yet allow for easy genetic manipulation and a virtually unlimited supply of cells. We demonstrate the utility of this system for dissection of host genetics during intracellular bacterial infection using two important human pathogens: *Listeria monocytogenes* and *Mycobacterium tuberculosis*.
DOI: https://doi.org/10.7554/eLife.45957.001

**\*For correspondence:**
jeff.cox@berkeley.edu

[†]These authors contributed equally to this work

**Competing interests:** The authors declare that no competing interests exist.

Although CRISPR/Cas9 technology has revolutionized our ability to manipulate genomes, cell-type specific barriers hinder genetic approaches to study many important mammalian cells and tissues. Macrophages are critical innate immune cells involved in tissue development, repair, and homeostasis as well as many microbial infections, but they are difficult to genetically manipulate via transfection or transduction, most likely due to sensitive circuits that sense foreign nucleic acid. Although it is possible to manipulate primary bone marrow derived macrophages (BMMs) using CRISPR/Cas9 technology (*Chu et al., 2016*), the low transduction and transfection efficiencies observed in these cells results in low editing efficiency or, if transductants can be selected, low cell numbers. These limited cell numbers preclude many biochemical and screening approaches that require many millions of cells. In addition, the short life-span of these cells does not allow for selection of individual mutant clones or subsequent genetic manipulations such as knockout of a second gene or protein overexpression. Because of these limitations many studies rely on either immortalized macrophage-like cell lines, which do not recapitulate important metabolic and inflammatory pathways of primary cells (*Andreu et al., 2017*), or the time-consuming process of generating transgenic or knockout mice from which to obtain BMMs.

To create an efficient and scalable system for effective genome editing in murine macrophages, we sought to build upon previous studies demonstrating that ectopic expression of an estrogen-regulated version of the homeobox transcription factor Hoxb8 (ER-Hoxb8) can immortalize macrophage progenitors that are self-renewing in the presence of β-estradiol (*Knoepfler et al., 2001*). Subsequent removal of the hormone activates normal differentiation of these cells into macrophages (*Wang et al., 2006*) (*Figure 1—figure supplement 1a*). We envisioned that deriving conditionally immortalized macrophage progenitors from Cas9-expressing mice (*Platt et al., 2014*) would allow us to perform gene editing in conditionally immortalized cells prior to differentiation, providing the opportunity to cryopreserve either bulk populations or individually-cloned mutant cells. Subsequent differentiation of edited progenitors into macrophages would then generate a theoretically

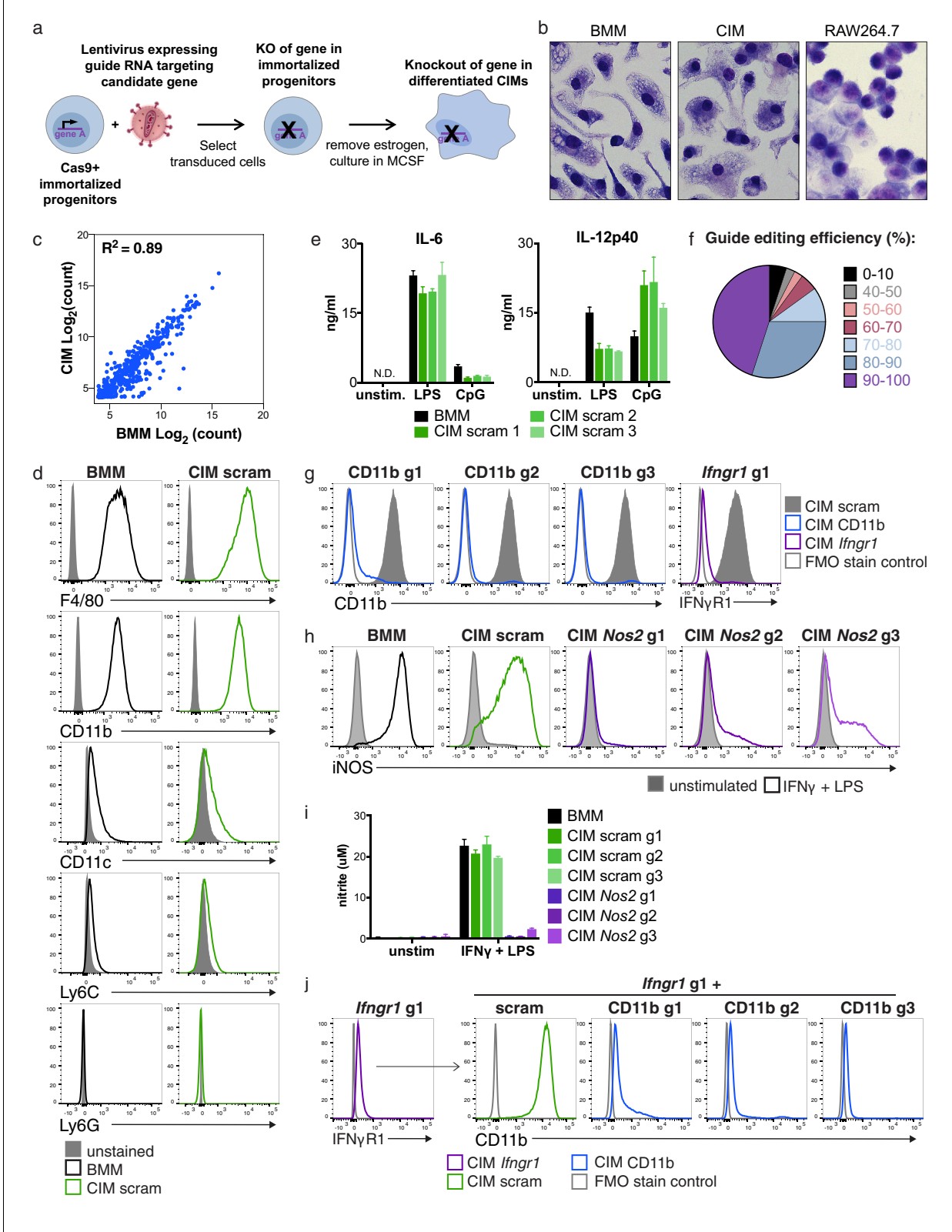

**Figure 1.** Cas9+ CIMs as a tractable system for genome-editing in macrophages. (a) Graphic overview of gene editing in Cas9+ CIMs. (b) BMMs (left panel), CIMs (middle), or RAW 264.7 cells (right) were visualized with Diff-Quick stain. (c) mRNA levels in BMMs and CIMs was quantified using a Nanostring nCounter. Data are representative of two independent experiments and are presented as log transformed normalized transcript counts of the average of technical duplicates from one experiment. (d) BMMs or CIMs transduced with a scramble guide (CIM scram) were analyzed by flow

*Figure 1 continued on next page*

*Figure 1 continued*

cytometry for expression of the indicated myeloid cell markers. Data are representative of three independent experiments. (**e**) IL-6 and IL-12p40 production by BMMs or CIM scram stimulated with the TLR4 ligand LPS or the TLR9 ligand CpG was measured by ELISA. N.D. – none detected. Data are representative of three independent experiments each performed in triplicate, mean ± SD are shown. (**f**) Genomic DNA from CIMs transduced with 40 guides targeting 17 genes was analyzed for genomic editing by TIDE analysis. (**g**) CIMs transduced with a scramble guide or guides targeting CD11b or *Ifngr1* were stained with the indicated antibodies. Fluorescence minus one (FMO) stained samples were used as controls. (**h and I**) BMMs or CIMs transduced with a scramble guide or guides targeting *Nos2* were stimulated overnight with LPS + IFNγ or left unstimulated and then analyzed by flow cytometry for expression of iNOS (**h**); nitric oxide production in cell-free supernatants was analyzed by Griess assay (**i**). Data are representative of two independent experiments each performed in triplicate, mean ± SD are shown. (**j**) CIM progenitors previously transduced with a lentivirus containing puromycin resistance and a guide targeting *Ifngr1* were subsequently transduced with lentivirus containing hygromycin resistance and scramble guide or guides targeting CD11b.

DOI: https://doi.org/10.7554/eLife.45957.002

The following figure supplement is available for figure 1:

**Figure supplement 1.** Re-selection of CIM progenitors that more closely resembled BMMs.

DOI: https://doi.org/10.7554/eLife.45957.003

unlimited supply of mutant macrophages for functional studies (*Figure 1a*). While others have recently taken a similar but limited approach using other lineages of Hoxb8 conditionally immortalized immune cells, we sought to fully establish the efficacy of Cas9[+] CIMs for robust gene editing and infection of macrophages (*Di Ceglie et al., 2017*; *Lee et al., 2017*; *MacDuff et al., 2018*).

To this end, we harvested hematopoietic stem cells from mice that constitutively express Cas9, infected them with lentivirus expressing the ER-Hoxb8 fusion protein and selected for cells that survived 3–4 weeks of culture in the presence of β-estradiol, as described by Wang, et. al. (*Wang et al., 2006*). These Cas9[+] macrophage progenitors grew robustly in suspension and, upon removal of β-estradiol and addition of MCSF, differentiated into adherent cells that expressed F4/80, a marker of macrophages (*Figure 1—figure supplement 1b and c*) (*Rosas et al., 2011*). However, in our initial studies we noted an unusual morphology of this initial population when compared to primary BMMs derived from wild-type C57BL/6 mice (*Figure 1—figure supplement 1b*); in addition, the cells expressed high levels of CD11c, a cell surface marker more closely associated with dendritic cells (*Figure 1—figure supplement 1c*). By re-selecting progenitors with high concentrations of the antibiotic G418, the resulting cell population more closely resembled BMMs morphologically (*Figure 1—figure supplement 1a and b*) and expressed lower levels of CD11c (*Figure 1—figure supplement 1c*). We speculate that high concentrations of G418 selected for progenitors with greater expression of the Hoxb8 fusion protein, skewing these cells towards a more uniform macrophage-committed progenitor population. Comparison of gross cellular morphology indicates that these Cas9[+] CIMs are much more similar to BMMs than transformed macrophage-like lines such as the frequently used RAW 264.7 (*Figure 1b*). To more globally compare BMMs and CIMs, we used Nanostring technology to measure mRNA levels of over 700 genes associated with myeloid innate immunity and compared the gene expression pattern between the two cell types. While the level of some of these mRNAs were different between BMMs and CIMs, the majority of transcripts were present at remarkably similar levels in both cell populations, consistent with the initial studies of similarly differentiated ER-HoxB8 cells, which demonstrated that these cells express many macrophage-specific genes but did not compare them directly to BMMs (*Figure 1c*) (*Wang et al., 2006*).

To determine if Cas9[+], gRNA-expressing CIMs possess functional macrophage phenotypes, we infected progenitors with one of three lentiviruses expressing non-targeting scramble gRNAs, selected transductants using puromycin, differentiated the resulting cells, and probed three macrophage characteristics to compare them with BMMs. First, flow cytometric analysis using key myeloid/lymphoid lineage markers revealed that CIMs were indistinguishable from BMMs, with high expression of the myeloid marker CD11b and macrophage marker F4/80, and low expression of CD11c, the monocyte marker Ly6C, and the neutrophil marker Ly6G (*Figure 1d*). Second, treatment with either lipopolysaccharide (LPS) (which engages TLR4) or CpG DNA (TLR9 agonist) induced the pro-inflammatory cytokines IL-6 and IL-12p40 from both CIMs and BMMs (*Figure 1e*). Although the levels were slightly different between the two cell types at these time points, it is clear that CIMs responded vigorously to innate immune stimulation upon engagement of pattern recognition receptors. Finally, CIMs and BMMs upregulated iNOS and generated similar levels of nitric oxide in

response to stimulation with LPS plus interferon-γ (IFNγ) (*Figure 1h and i*). Based on these results, we conclude that transduced Cas9+ CIMs share many of the characteristics of BMMs.

To broadly assess genome editing efficacy in Cas9+ CIMs, we introduced 40 individual gRNAs targeting a total of 17 genes into these cells, selected for puromycin resistant cells, and assessed genome editing in the resulting populations by PCR amplification of the target sites and sequence analysis using Tracking Indels by Decomposition (TIDE) analysis (*Brinkman et al., 2014*). We found that over 80% of these gRNA transductions resulted in at least 70% editing of the target gene after differentiation, with the majority achieving 80–100% editing efficiency (*Figure 1f*). For all 11 genes independently targeted with three distinct gRNAs, we obtained >80% editing from at least one guide RNA. To test whether genome editing in these polyclonal populations resulted in significant differences in protein levels, we examined surface expression of CD11b and IFNγ receptor after transduction with gRNAs targeting CD11b or *Ifngr1* and found that over 90% of CIMs had substantially reduced expression of the targeted protein as assessed by flow cytometry (*Figure 1g*). Likewise, transduction with lentiviruses encoding one of three separate guides targeting *Nos2*, the gene encoding the inducible nitric oxide synthase (iNOS), effectively blocked iNOS expression (*Figure 1h*) and production of NO (*Figure 1i*) in response to LPS + IFNγ in all three CIM populations. Although gene-to-gene variability in CRISPR/Cas9-mediated targeting efficiency will certainly exist, taken together our data indicate that independent transduction with three distinct gRNAs per gene in Cas9+ CIMs will disrupt the majority of genes efficiently enough to produce phenotypically relevant changes in gene expression using bulk populations of transduced cells.

Another major advantage of the Cas9+ CIM system is its potential for studying genetic interactions by generating double gene knockouts in the immortalized progenitor state prior to differentiation. To this end, we transduced Cas9+ macrophage progenitors with a lentivirus containing puromycin resistance and targeting *Ifngr1* and subsequently transduced the resulting population with lentiviruses containing hygromycin resistance and expressing one of three CD11b or scramble guides. After differentiation of these doubly-resistant populations, over 90% of CIMs lacked both IFNγR1 and CD11b protein expression as assessed by flow cytometry (*Figure 1j*). This iterative approach provides a robust way to generate double knockouts, enabling a simple methodology to elucidate synthetic interactions. Overall, our results establish Cas9+ CIMs as an efficient and robust model for genome editing in macrophages with distinct advantages to standard transformed macrophage-like cell lines.

To more specifically test whether CIMs are effective for studying the antimicrobial functions of macrophages, we assessed the host and microbe requirements of two well-established bacterial pathogens that naturally infect macrophages, *Listeria monocytogenes* and *Mycobacterium tuberculosis*. First, *L. monocytogenes* incubated with either BMMs or CIMs were effectively phagocytosed, replicated robustly, and spread to neighboring macrophages (*Figure 2a and c*). After phagocytosis, *L. monocytogenes* ruptures phagosomal membranes using its secreted pore-forming toxin listeriolysin O (LLO, encoded by the *hly* gene) in order to grow in the cytosol, and expresses ActA to hijack the host actin polymerization machinery in order to propel itself through the cytosol and mediate spread to neighboring cells (*Portnoy et al., 2002*). Importantly, *L. monocytogenes* lacking LLO (Δ*hly*) were unable to grow in BMMs or CIMs, and bacteria lacking ActA (Δ*actA*) failed to recruit actin and spread (*Figure 2b*). Thus, the major bacterial virulence determinants required for *L. monocytogenes* infection are the same for BMMs and CIMs.

Targeting microbes for destruction via the host autophagy pathway, a process termed xenophagy, is a major mechanism employed by primary macrophages for controlling intracellular pathogens, including *L. monocytogenes* (*Gomes and Dikic, 2014*). To determine whether CIMs are capable of restricting *L. monocytogenes* through autophagy targeting, we used a previously characterized bacterial mutant strain lacking three key virulence factors (ActA, PlcA, PlcB) required for microbial evasion of autophagy in BMMs (*Mitchell et al., 2018*). Kinetic growth assays revealed that this autophagy-sensitive *L. monocytogenes* strain, and the LLO-defective *L. monocytogenes* mutant, are both significantly attenuated in both BMMs and CIMs (*Figure 2c*). Bacterial burden associated with all three strains of *L. monocytogenes* was somewhat reduced in CIMs compared to BMMs, which we speculate is due to enhanced early killing by the phagolysosomal pathway. Similar results have been observed in peritoneal macrophages, which more effectively engage LC3-associated phagocytosis and are more microbicidal than BMMs against *L. monocytogenes* early in infection (*Gluschko et al., 2018*; *Herskovits et al., 2007*). To examine host genetics of microbial autophagy

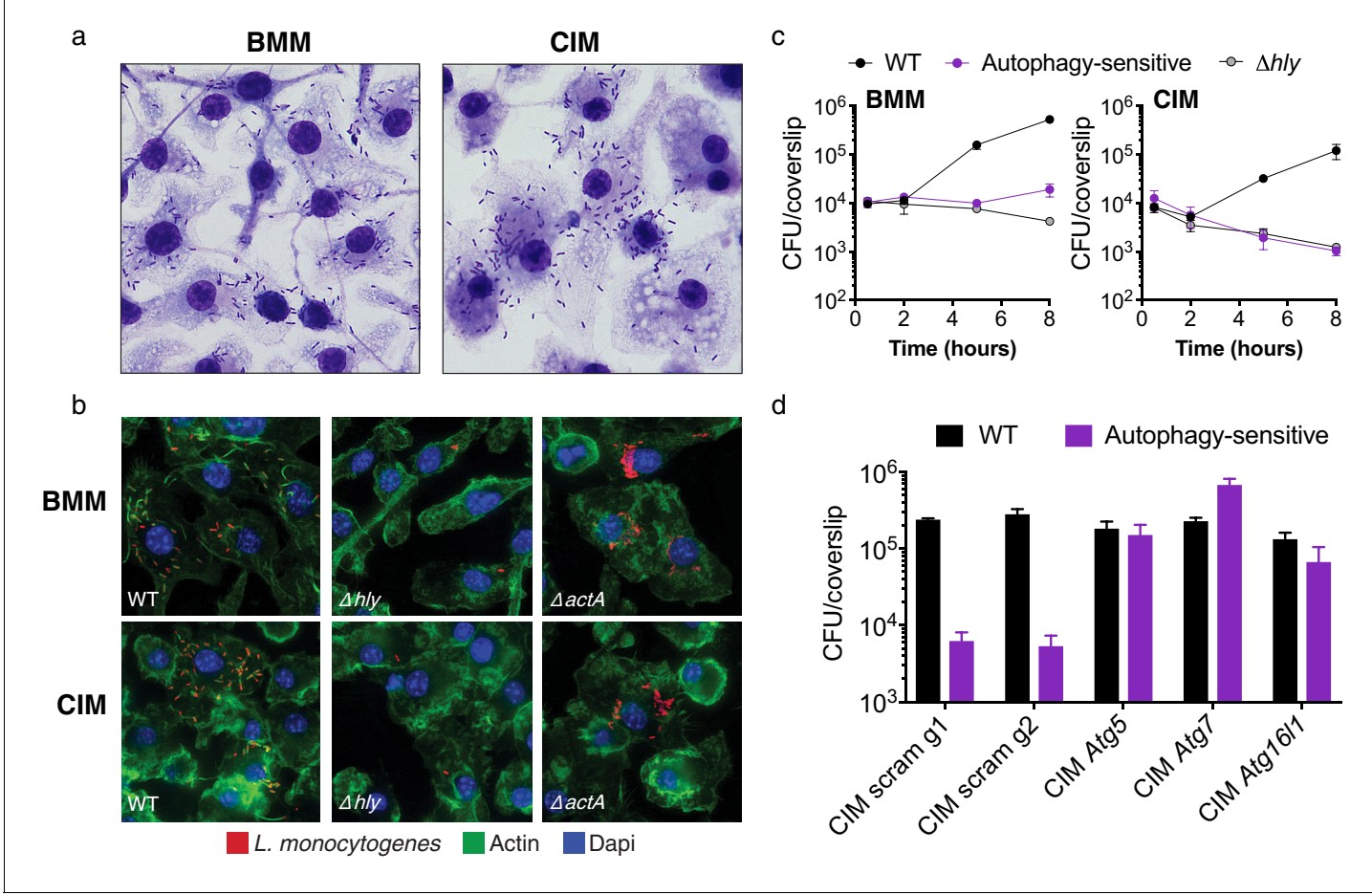

**Figure 2.** Cas9[+] CIMs as an in vitro model for *Listeria monocytogenes* infection. (**a**) BMMs (left panel) or CIMs (right) were infected with WT *Listeria monocytogenes* at MOI = 0.25 and monolayers were visualized with Diff-Quick stain at 8 hr post-infection. Data are representative of two independent experiments. (**b**) BMMs (top three panels) or CIMs (lower panels) were infected for 5 hr with three strains of *L. monocytogenes:* WT, Δ*hly*, and Δ*actA* at MOI = 1.5. Nuclei shown in blue, bacterial cells in red, and F-actin in green. (**c**) BMMs (left panel) or CIMs transduced with a scramble gRNA were infected with three strains of *L. monocytogenes:* WT, an autophagy-sensitive strain that lacks ActA, PlcA and PlcB (*Mitchell et al., 2018*) and Δ*hly* at MOI = 0.25. Bacterial densities were enumerated by CFU at *t* = 0.5 h and indicated hours post-infection. Data are representative of two independent experiments each performed in triplicate, mean ± SD are shown. (**d**) CIMs transduced with the indicated gRNAs were infected with WT *L. monocytogenes* or the autophagy-sensitive *L. monocytogenes* mutant at MOI = 0.75 and bacterial density was enumerated by CFU at *t* = 8 hr. Data are representative of two independent experiments each performed in triplicate, mean ± SD are shown.

DOI: https://doi.org/10.7554/eLife.45957.004

targeting, we generated CIMs deficient in one of several key components of the autophagy pathway and subsequently infected them with WT *L. monocytogenes* and the autophagy-sensitive strain. Abrogation of the autophagy pathway through knockout of *Atg5*, *Atg7*, or *Atg16l1* restored replication of the autophagy-sensitive *L. monocytogenes* strain relative to scramble CIM controls (*Figure 2d*), demonstrating that xenophagic targeting is intact and functional in CIMs.

*Mycobacterium tuberculosis* is a major human pathogen and its ability to replicate and persist within macrophages, as well as to interact with the adaptive immune system, is central to its long-term pathogenic strategy (*Upadhyay et al., 2018*). Its most important virulence determinant is the type VII protein secretion system termed ESX-1, which is critical for growth in macrophages and influences the innate immune responses of these cells (*Stanley et al., 2003*). To assess CIMs as a model for studying *M. tuberculosis* invasion of macrophages, we infected BMMs and CIMs with auto-luminescent *M. tuberculosis* strains and quantified bacterial replication over a five-day time course. We found that wild-type *M. tuberculosis* grew robustly in CIMs, with kinetics comparable to that observed in BMMs (*Figure 3a*). CIM monolayers began to lose integrity six days after infection

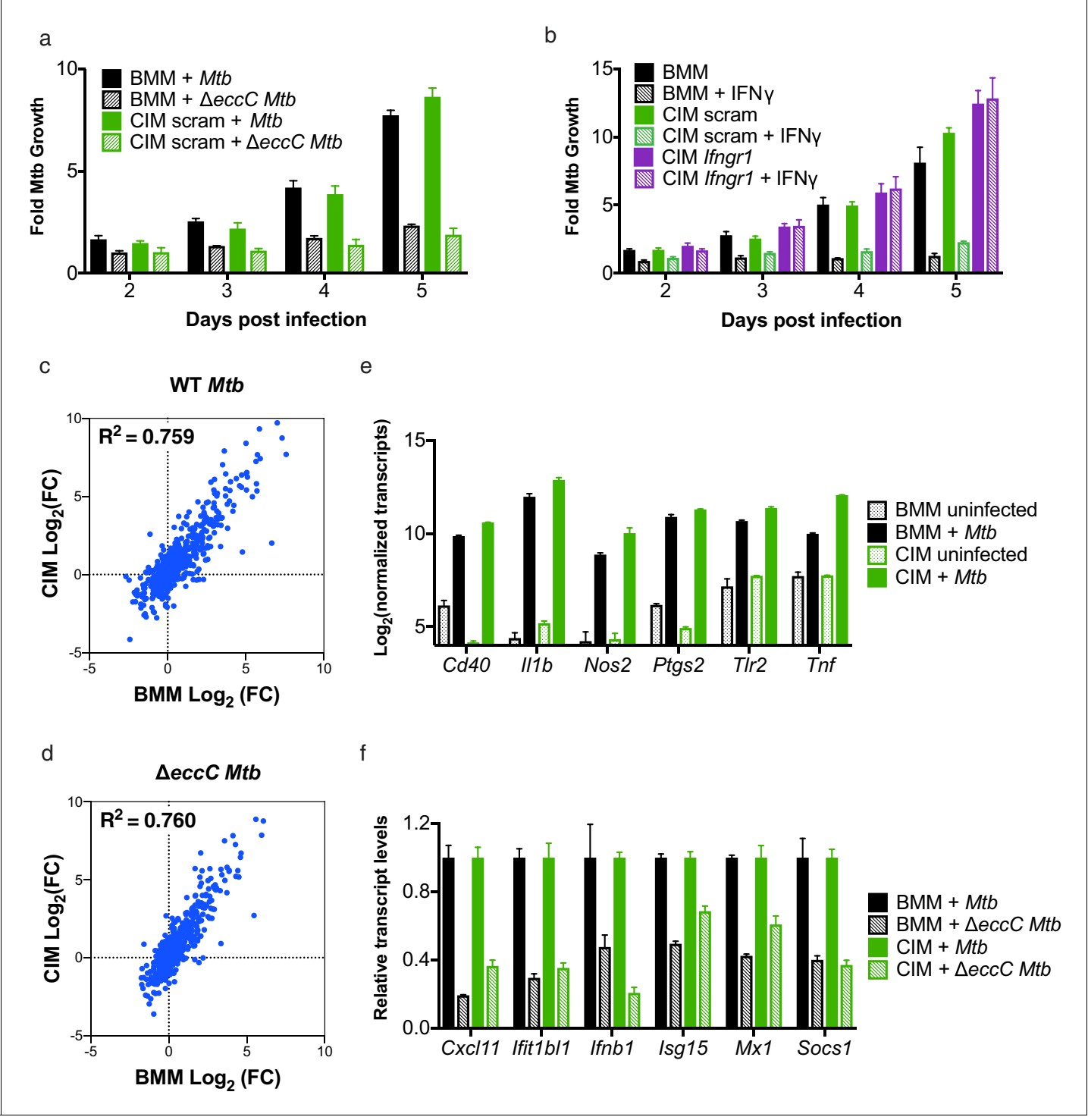

**Figure 3.** Cas9[+] CIMs as an in vitro model for *Mycobacterium tuberculosis* infection. (**a**) Luminescent bacterial growth assay. BMMs (black) vs. CIMs transduced with a scramble guide (green) were infected with *M. tuberculosis*-Erdman (solid bars) or a Δ*eccC M. tuberculosis*-Erdman strain (patterned bars) carrying the *luxCDABE* reporter operon at MOI = 0.5. Data are representative of two independent experiments each performed in triplicate, mean ± SD are shown. (**b**) Luminescent bacterial growth assay as in a, BMMs (black), CIMs transduced with a scramble guide (green), or CIMs transduced with a guide targeting *Ifngr* (purple) were infected with *M. tuberculosis*-Erdman Lux. Patterned bars indicate conditions where the different cell types were treated with IFNγ prior to and throughout infection. Fold-change in luminescence is reported as *M. tuberculosis* growth relative to *t* = 0 for each condition. Data are representative of four independent experiments each performed in triplicate, mean ± SD are shown. (**c–f**) mRNA levels in BMMs and CIMs, either uninfected or infected with WT (**c**) or Δ*eccC M. tuberculosis* (**d**) at MOI = 5 at 6 hr post infection were quantified using a

*Figure 3 continued on next page*

*Figure 3 continued*

Nanostring nCounter. Data are representative of two independent experiments. Source data is available as *Figure 3—source data 1*. **c** and (**d**) Data are presented as fold changes of infected/uninfected values of the average of technical duplicates from one experiment. (**e**) Log-transformed, normalized transcript counts for the indicated genes obtained from BMMs (black) vs. CIMs (green) before (patterned bars) or after infection with WT *M. tuberculosis* (solid bars). (**f**) Transcript counts for the indicated genes obtained from BMMs (black) vs. CIMs (green) infected with WT *M. tuberculosis* (solid bars) or *M. tuberculosis ΔeccC* (patterned bars). Transcripts were normalized to counts in WT *M. tuberculosis* infected macrophages.

DOI: https://doi.org/10.7554/eLife.45957.005

The following source data and figure supplement are available for figure 3:

**Source data 1.** mRNA levels in BMMs and CIMs, uninfected or infected with *M.*

DOI: https://doi.org/10.7554/eLife.45957.007

**Figure supplement 1.** Weak correlation between gene induction by RAW 264.7 cells and BMMs or CIMs in response to *M. tuberculosis*.

DOI: https://doi.org/10.7554/eLife.45957.006

as the macrophages initiated cell death, a hallmark of *M. tuberculosis* infection (*Chen et al., 2008*), which occurred slightly earlier than in BMM monolayers (data not shown). To determine whether ESX-1 is required for *M. tuberculosis* replication in CIMs, we also infected both macrophage types with an auto-luminescent *M. tuberculosis* strain lacking the core ATPase for the secretion system, *eccC*, which results in complete loss of ESX-1 secretion (*Rosenberg et al., 2015*). As expected, these mutant bacteria were unable to replicate in BMMs or CIMs, indicating that the requirements for *M. tuberculosis* replication are the same for both types of macrophages (*Figure 3a*).

IFNγ-mediated activation of macrophage antimicrobial defenses during in vivo infection is critical for host resistance to *M. tuberculosis* infection, and ex vivo treatment of cultured macrophages with this cytokine induces BMMs to inhibit bacterial replication (*Flynn et al., 1993*). Importantly, addition of IFNγ to CIMs transduced with lentivirus encoding scramble gRNA restricted *M. tuberculosis* growth to a similar extent as IFNγ treated BMMs (*Figure 3b*). Abrogating IFNγ signaling through knockout of *Ifngr1* (*Figure 1g*) restored replication of *M. tuberculosis* in IFNγ treated CIMs (*Figure 3b*). These data indicate that CIMs can provide the environmental conditions required for *M. tuberculosis* growth, but also have the inducible capacity to restrict bacterial replication.

Macrophage-like cell lines and BMMs have been demonstrated to induce markedly different responses after *M. tuberculosis* infection (*Andreu et al., 2017*). To determine how closely CIMs resemble BMMs during infection with *M. tuberculosis*, we globally compared gene expression of BMMs and CIMs during infection with either a virulent WT *M. tuberculosis* Erdman strain or *M. tuberculosis ΔeccC* lacking a functional ESX-1 secretion system using the same 700 myeloid gene panel as described above. There was strong correlation between the genes induced by BMMs and CIMs in response to both WT and *ΔeccC M. tuberculosis* (*Figure 3c and d*). Lentiviral transduction with a non-targeting gRNA did not significantly alter the response of CIMs to *M. tuberculosis* (*Figure 3—figure supplement 1a*). In contrast to the comparison between BMMs and CIMs, there was weak correlation between the genes induced by RAW 264.7 cells and BMMs or CIMs in response to *M. tuberculosis* (*Figure 3—figure supplement 1b and c*). Both BMMs and CIMs activated genes known to be involved in *M. tuberculosis* infection, including *Cd40*, *Il1b*, *Nos2*, *Ptgs2*, *Tlr2*, and *Tnf* (*Andreu et al., 2017*) (*Figure 3e*). As previously demonstrated in BMMs, we observed that many interferon-stimulated genes were more highly induced upon infection with WT *M. tuberculosis* than *M. tuberculosis* lacking ESX-1 in both BMMs and CIMs (*Manzanillo et al., 2012*; *Stanley et al., 2007*) (*Figure 3f*). Overall, these results establish CIMs as a suitable model for *M. tuberculosis* infection of macrophages.

Our results establish Cas9⁺ CIMs as a powerful ex vivo model of macrophage biology and represent a major technological advance in our ability to capitalize on powerful genome editing capabilities in this notoriously refractory cell type. We expect that the methods we have developed will lay the foundation for investigations exploring new pathways that mediate the many roles of this important cell type in mammalian biology (*Wynn et al., 2013*). Indeed, while we show that Cas9⁺ CIMs are a flexible tool for studying some important aspects of innate immunity and inflammation, they are also likely to be useful for unbiased screening of CRISPR libraries to identify pathways important for other macrophage functions including tissue homeostasis and metabolic reprogramming. Because Cas9⁺ CIMs can give rise to an unlimited number of mutant macrophages, this scalable system can generate genetically defined mutant macrophages for use in a wide range of applications,

from in vitro biochemical approaches to in vivo adoptive transfer studies of mice (*Redecke et al., 2013*; *Wiesmeier et al., 2016*). Furthermore, using an iterative genome-editing approach, Cas9+-CIMs allow for facile exploration of large-scale genetic interactions. Thus, this system represents a significant technological advance that will likely promote the study of macrophages and their myriad functions.

# Materials and methods

## Key resources table

| Reagent type (species) or resource | Designation | Source or reference | Identifiers | Additional information |
|---|---|---|---|---|
| Strain, strain background (*M. musculus*) | WT C57BL/6J | Jackson Laboratory | Stock No: 000664 | |
| Genetic reagent (*M. musculus*) | Rosa26-Cas9 knockin mouse | Jackson Laboratory | Stock No: 026179 | *Platt et al., 2014* |
| Recombinant DNA reagent | ER-Hoxb8-MSCV-Neo | *Wang et al., 2006* | | |
| Recombinant DNA reagent | pLentiGuide-Puro | Addgene | 52963 | |
| Recombinant DNA reagent | lenti-sgRNA hygro | Addgene | 104991 | |
| Strain, strain background (*M. tuberculosis*) | Erdman strain | | BEI Cat # NR-15404 | |
| Genetic reagent (*M. tuberculosis*) | ΔeccC (Erdman, ΔeccCa1-ΔeccCb1) | *Rosenberg et al., 2015* | | |
| Genetic reagent (*M. tuberculosis*) | Lux (Erdman expressing the *luxCDABE* operon) | *Braverman et al., 2016* | | |
| Genetic reagent (*M. tuberculosis*) | ΔeccC Lux | *Penn et al., 2018* | | |
| Strain, strain background (*L. monocytogenes*) | 10403S strain | PMID: 24667708 | | |
| Genetic reagent (*L. monocytogenes*) | ΔactA | PMID: 10931865 | DP-L3078 | |
| Genetic reagent (*L. monocytogenes*) | Δhly | PMID: 7960143 | DP-L2161 | |
| Genetic reagent (*L. monocytogenes*) | ΔactA PlcA$^{H86A}$ PlcB$^{H69G}$ | *Mitchell et al., 2018* | DP-L6586 | |
| Commercial assay or kit | nCounter Mouse Myeloid Innate Immunity Panel | NanoString | | |

## Cell culture

Bone marrow derived macrophages (BMMs) were generated from the femurs and tibias from wild-type C57BL/6 (The Jackson Laboratory) mice that were 8–12 weeks old. Conditionally-immortalized macrophages (CIMs) were derived from bone marrow cells from a 5-FU treated male Cas9+ mouse (*Platt et al., 2014*). Hoxb8 immortalized cells were generated as previously described (*Wang et al., 2006*) with the modification that progenitors in bone marrow were enriched by negative depletion; cells expressing CD11b, Ter119, B220, CD5, CD19, and Gr-1 were depleted using biotinylated anti-bodies and streptavidin coated dynabeads. For re-selection of CIM progenitors, cells were cultured in 10 mg/ml G418 for 7 days. All mice were housed in specific-pathogen free conditions and treated using procedures described in animal care protocols approved by the Institutional Animal Care and Use Committee of UC Berkeley.

Progenitor CIMs prior to differentiation were maintained in RPMI (Gibco) supplemented with 10% FBS, 2% GM-CSF supernatant produced by a B16 murine melanoma cell line, 2 mM L-glutamine, 1 mM sodium pyruvate, 10 mM HEPES, 43 uM β-mercaptoethanol, and 2 uM β-estradiol (Sigma #E2758). Progenitor CIMs were maintained in suspension in non-treated tissue culture treated flasks at densities below 500,000 cells/ml before removal of β-estradiol and differentiation. BMMs and differentiated CIM macrophages were cultured in macrophage media: DMEM (Gibco) supplemented with 10% FBS, 10% M-CSF supernatant produced by 3T3-MCSF cells as previously described, 2 mM L-glutamine (Gibco), and 1 mM sodium pyruvate (Gibco). RAW 264.7 cells were cultured in DMEM (Gibco) supplemented with 10% FBS, 2 mM L-glutamine (Gibco), and 20 mM HEPES. To differentiate progenitor CIMs into macrophages, cells were washed twice in PBS + 1% FBS to fully remove β-estradiol, resuspended in complete macrophage media, and seeded onto non-treated 15 cm tissue culture plates at $5.0 \times 10^6$ cells/plate in 20 ml of media. Differentiating CIM macrophages were given an additional 10 ml of macrophage media on days 3 and 6 post-differentiation, and terminal assays were performed at day 9 or 10 post-differentiation.

## Transfection, transduction, genotyping, and sequence analysis

CRISPR guide sequences targeting genes of interest were selected from the murine Brie guide library. Oligonucleotides encoding the chosen gRNAs (see *Supplementary file 1*) were cloned into pLentiGuide-Puro (Addgene #52963) or lenti-gRNA hygro (Addgene #104991), and verified by sequencing using the human U6 sequencing primer. 293 T cells were co-transfected with pLenti-Guide-Puro, psPAX2, and pMD2.G using Lipofectamine and Optimem according to manufacturer's guidelines to generate lentiviral particles for trandsuction into Cas9-expressing CIM progenitors. For optimal transduction of Cas9-expressing CIM progenitors, $5.0 \times 10^5$ cells/well in a 6-well plate were spinfected at 1000xg for 2 hr at 32°C in the presence of 10 µg/ml protamine sulfate. Two days post-transduction, 12 µg/ml puromycin was added to cells, and cells were selected in puromycin for 4 days. Puromycin-resistant cells were maintained as polyclonal populations and total gDNA was extracted using DNeasy Blood and Tissue kit (Qiagen). To generate double knockout progenitors, puromycin-resistant progenitors that had previously been transduced with pLentiGuide-Puro were transduced with lenti-sgRNA hygro and selected using 250 µg/ml hygromycin for 8 days. The genomic sites encompassing targeted guide regions were amplified by PCR using iProof polymerase (Bio-Rad) and sequenced, and population level genome editing was estimated using the TIDE webtool (https://tide.deskgen.com/) as originally described (*Brinkman et al., 2014*). Guide genome editing efficiencies displayed were a combination of data from CIMs before and after G418 selection, with no notable difference in editing efficiency caused by G418 selection.

## *Mycobacterium tuberculosis* infection

CIMs or BMMs were seeded at 60,000 cells per well onto white, clear-bottom CellBind 96-well plates (Corning) or 12-well TC-treated plates (Corning) in macrophage media one or two days prior to infection. For pre-treatment of macrophages with IFN-γ, post-seeding cell culture media was switched to media with 1.5 ng/µl of recombinant murine IFN-γ (Peprotech) 12–18 hr prior to infection, and activated cells were subsequently cultured in IFN-γ containing media throughout infection.

Macrophages were infected with *M. tuberculosis* as previously described (*Penn et al., 2018*). The *M. tuberculosis* strain Erdman the ΔeccC strain made in the Erdman background, or the WT or ΔeccC strain made in the Erdman background expressing the *luxCDABE* operon (*Braverman et al., 2016*; *Penn et al., 2018*; *Rosenberg et al., 2015*) were used for all infections. All Mtb strains were cultured in 7H9 liquid media (BD) supplemented with 10% Middlebrook OADC (Sigma), 0.5% glycerol, 0.05% Tween-80 in roller bottles at 37°C. Briefly, mid-log *M. tuberculosis* cultures were washed twice with PBS, gently sonicated to disperse clumps, and resuspended in phagocytosis infection media (DMEM supplemented with 5% horse serum and 5% FBS). For luminescence assays macrophages were infected in at least triplicate wells by removing media from cells, and monolayers were overlaid with the bacterial suspensions in phagocytosis media then incubated at 37°C for 4 hr, after which infection media was removed and fresh macrophage media was added. Bacterial luminescence signal was measured at 32°C at the time of infection and every day starting 48 hr post-infection after daily media changes. All growth measurements are normalized to day 0 luminescence readings for each infected well and are presented as fold change in luminescence compared to day

0. Cells were infected at a multiplicity of infection (MOI) of 5 for RNA analysis and 0.5 for assays monitoring bacterial growth.

### *Listeria monocytogenes* infection

The WT 10403S, Δ*actA* (DP-L3078), Δ*hly* (DP-L2161) and the autophagy-sensitive Δ*actA* PlcA$^{H86A}$ PlcB$^{H69G}$ (DP-L6586) strains (*Mitchell et al., 2018*) (*Cheng et al., 2018*) were grown overnight in brain heart infusion (BHI) medium at 30°C before each experiment. Infections of macrophages were performed as previously described (*Cheng et al., 2018*; *Mitchell et al., 2018*) with few modifications. BMMs and CIMs were seeded on round 12 mm coverslips (Fisher Scientific, Hampton, NH, USA) and incubated overnight at 37°C and 5% CO$_2$. Coverslips were incubated with a solution 0.1% gelatin for 1 hr at 37°C and washed twice with PBS prior seeding CIMs. Macrophages were infected at various MOIs, washed at 0.5 hr post-infection and further incubated in fresh medium. At 1 hr post-infection, 50 μg/ml of gentamicin was added to the medium in order to kill extracellular bacteria. MOIs and incubation times are indicated in figure legends.

### Flow cytometry

Cells were fixed with Fix/Perm buffer (BD) for 20 min at 4°C. All stains were carried out in PBS containing 2% FBS (v/v) including anti-CD16/32 Fc blocking antibody (clone 93, BioLegend). Cells were stained for 20 min at 4°C with antibodies against F4/80 (clone BM8, Tonbo Biosciences), CD11b (clone M1/70, Tonbo Biosciences), CD11c (clone N418, Tonbo Biosciences), Ly6C (clone HK 1.4, BioLegend), Ly6G (clone 1A8, Tonbo Biosciences), IFNγR1 (clone 200, eBioscience) or iNos (clone CXNFT, eBioscience). All cells were analyzed on an LSR Fortessa (BD Biosciences), and data was analyzed with FlowJo.

### TLR ligand and IFNγ stimulations

Cells were seeded at 60,000 cells per well in 96-well CellBind plates (Corning) in macrophage media the day prior to stimulation. For cytokine analysis cells were stimulated with 1 μg/ml LPS (InvivoGen) or 1 uM CpG-1668 (InvivoGen, Tlrl-1668) for 16–20 hr. Cell-free supernatants were harvested and frozen at −20°C prior to analysis. For Griess assay analysis, cells were stimulated with 100 ng/ml LPS and 5 ng/ml IFNγ. After 24 hr cell-free supernatant was harvested and immediately analyzed.

### ELISA

Cell-free supernatants from stimulated cells were analyzed by ELISA to enumerate cytokine production. Supernatants were absorbed onto a 96-well Nunc MaxiSorp flat-bottom plate (44-2404-21, Invitrogen) coated with the following capture antibodies diluted in 0.1M sodium phosphate buffer pH 8.0 and incubated overnight at 4°C: IL-6 (clone MP5-20F3, 554400, BD Biosciences, 1 μg/ml) and IL-12 p40 (clone C15.6, 14-7125-81, eBioscience, 1 μg/ml). The following day, plates were washed 3x with PBS + 0.05% Tween-20 and blocked with PBS + 1% BSA for 4 hr at RT. Recombinant cytokines for standard measurements were serially diluted in PBS/BSA. IL-6 (406 ML-025, R and D) IL-12 p40 (34-8321-63, eBioscience) and experimental supernatants were incubated overnight at 4°C. After 3X washes in PBS/BSA, biotin-conjugated sandwich antibodies were incubated on supernatants and standards (IL-6, MP5-32C11, 554402, BD Biosciences and p40 C17.8, 13-7123-85, eBiosciences) and detected by Streptavidin-HRP (BD Biosciences, 1:3000). For the development step, ELISA was incubated with OPD (Sigma) and 30% hydrogen peroxide, followed by incubation with 3M HCl. Absorbance at 490 nm was read on an Infinite M200 Tecan plate reader.

### Griess assay

Cell-free supernatants from stimulated cells were analyzed by Griess reaction to detect nitrite as a proxy for NO production. A solution of 0.2% napthylethylenediamine dihydrochloride was mixed 1:1 with a 2% sulfanilamide/4% phosphoric acid solution. 100 μl of this solution was mixed with 100 μl of sample supernatant and absorbance at 546 nm was immediately measured. Nitrite concentrations were determined using a standard curve of sodium nitrite.

## Immunofluorescence microscopy and Diff-Quick staining

Coverslips seeded with macrophages were left uninfected or infected with *L. monocytogenes*, stained with a Diff-Quick stain set (Dade Behring, Deerfield, IL, USA), air-dried and mounted on glass slides using a drop of Permount (Fisher Scientific). Immunofluorescence staining of macrophages infected with *L. monocytogenes* was performed using a polyclonal rabbit antisera that recognizes *L. monocytogenes* (BD Biosciences, San Jose, CA, USA), a rhodamine red-X goat anti-rabbit IgG (Invitrogen, Carlsbad, CA, USA), phalloidin AlexaFluor-647 (Invitrogen) and ProLong Gold antifade reagent containing 4′6,-diamidino-2-phenylindole (Invitrogen), as previously described (*Cheng et al., 2018*). All images were acquired with a KEYENCE BZ-X710 fluorescent microscope using a 100 × objective and post-treated using the haze reduction function of the BZ-X analyser software. Images showed in *Figure 2B* were obtained by staking 10 layers covering 5 µm of thickness and pseudo-colored.

## mRNA analysis using nanostring Ncounter

Total RNA was isolated using TRIzol (Fisher) and the PureLink RNA Mini Kit (12183018A, Ambion), NEB DNase treated, and purified using RNA clean and concentrator columns (Zymo). RNA was analyzed using the mouse myeloid innate immunity panel of the NanoString nCounter Analysis System (NanoString Technologies). Raw counts of samples were normalized according to the manufacturer's recommendations using reference genes as internal controls (*Sdha, Oaz1, Rpl19, Edc3, Sap130, Hdac3, Polr2a, Ppia, Gusb, Tbp, Sf3a3, and Abcf1*). Background threshold was set to the geometric mean of the negative controls. Normalization was performed using nSolver Analysis Software v4. Normalized transcript counts are shown in *Figure 3—source data 1*.

## Statistics

Statistical analysis of data was performed using GraphPad Prism software (Graphpad, San Diego, CA). Results are reported as the mean $\pm$ SD. Pearson correlation coefficients and $R^2$ values for scatter plots of mRNA expression were determined using log transformed data.

# Acknowledgements

We thank Nevan J Krogan as well as members of the Cox, Barton, and Stanley labs for helpful discussions and advice. This work was supported by NIH Grants P01 AI063302 (JSC and GMB), U19 AI135990 (JSC and GMB), DP1 AI124619 (JSC), R01 AI072429 (GMB). Allison Roberts is an Open Philanthropy Fellow of the Life Sciences Research Foundation.

# Additional information

### Funding

| Funder | Grant reference number | Author |
| --- | --- | --- |
| National Institutes of Health | P01AI063302 | Gregory M Barton<br>Jeffery S Cox |
| National Institutes of Health | U19AI135990 | Gregory M Barton<br>Jeffery S Cox |
| National Institutes of Health | DP1AI24619 | Jeffery S Cox |
| National Institutes of Health | R01AI072429 | Gregory M Barton |
| Life Sciences Research Foundation | Open Philanthropy Fellow | Allison W Roberts |

The funders had no role in study design, data collection and interpretation, or the decision to submit the work for publication.

### Author contributions

Allison W Roberts, Conceptualization, Formal analysis, Funding acquisition, Validation, Investigation, Methodology, Writing—original draft, Writing—review and editing; Lauren M Popov,

Conceptualization, Formal analysis, Validation, Investigation, Methodology, Writing—original draft, Writing—review and editing; Gabriel Mitchell, Investigation, Writing—review and editing; Krystal L Ching, Resources, Investigation; Daniel J Licht, Guillaume Golovkine, Resources, Writing—review and editing; Gregory M Barton, Conceptualization, Resources, Supervision, Funding acquisition; Jeffery S Cox, Conceptualization, Supervision, Funding acquisition, Writing—original draft, Project administration, Writing—review and editing

## Author ORCIDs

Allison W Roberts (iD) https://orcid.org/0000-0001-6681-4144
Gabriel Mitchell (iD) http://orcid.org/0000-0001-7977-5316
Krystal L Ching (iD) http://orcid.org/0000-0002-1181-0119
Gregory M Barton (iD) http://orcid.org/0000-0002-3793-0100
Jeffery S Cox (iD) https://orcid.org/0000-0002-5061-6618

## Ethics

Animal experimentation: This study was performed in strict accordance with the recommendations in the Guide for the Care and Use of Laboratory Animals of the National Institutes of Health. All mice were treated using procedures described in animal care protocols approved by the Institutional Animal Care and Use Committee of UC Berkeley (Protocol # AUP-2015-11-8096-1).

## Decision letter and Author response

Decision letter https://doi.org/10.7554/eLife.45957.011
Author response https://doi.org/10.7554/eLife.45957.012

## Additional files

### Supplementary files

• Supplementary file 1. List of gRNA sequences and primers used to determine editing efficiency.
DOI: https://doi.org/10.7554/eLife.45957.008

• Transparent reporting form
DOI: https://doi.org/10.7554/eLife.45957.009

### Data availability

All data generated or analyzed during this study are included in the manuscript and supporting files. Source data files have been provided for Figure 3.

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
