## [Decision Letter]

Thank you for submitting your article "Cas9^+^ Conditionally-Immortalized Macrophages: a tool for bacterial pathogenesis and beyond" for consideration by *eLife*. Your article has been reviewed by four peer reviewers, one of whom is a member of our Board of Reviewing Editors, and the evaluation has been overseen by Wendy Garrett as the Senior Editor. The reviewers have opted to remain anonymous.

The reviewers have discussed the reviews with one another and the Reviewing Editor has drafted this decision to help you prepare a revised submission.

Summary:

The authors set out to create an approach that will yield a cell type that is close to primary bone marrow-derived macrophages (BMMs), and that can be (i) produced in abundance ex vivo and (ii) be genetically manipulated through CRISPR-CAS9 to modulate specific pathways. The cell type produced are referred to as Conditionally-Immortalized Macrophages (CIM).

Approach: They use a regulated version of the homeobox transcription factor Hoxb8 (ER-Hoxb8) in conditionally immortalized macrophage progenitors from Cas9-expressing mice, with the idea that differentiation of these can lead to unlimited supply of macrophages. By comparing the gross morphology of the resulting cells, the authors demonstrate similarity to BMMs, an observation further confirmed using NanoString technology to measure mRNA of

700 genes associated with myeloid innate immunity. The functionality (determined by measuring induction of pro-inflammatory cytokines IL-6 and IL-12p40 and measuring iNOS activity) and similarity of CIMs is maintained when these cells are transfected with the lentivirus system used to deliver short guide RNAs for gene knockdown. Next, to determine the capacity of these CIMs for gene editing the authors introduced 40 individual short guide RNAs that target 17 genes and assessed knockdown using Tracking Indels by Decomposition (TIDE) analysis. In this set of experiments, they obtained >80% gene knockdown from at least one guide RNA. Following this, they also develop a system to make double gene knockdowns in the CIM model.

Validation: To validate the utility of the CIMs produced by the authors, they use two bacterial systems, *Listeria monocytogenes* and *Mycobacterium tuberculosis* (with BMMs the comparator).

*L. monocytogenes*: Here they monitored phagocytosis and replication, which were comparable between the two cell types (CIMs and BMMs). They also monitored secretion of the listeriolysin O toxin and the ability of the bacterium to make actin comet tails, these bugs move around the cytoplasm by recruiting actin, through a protein called ActA. In addition, using mutants of *L. monocytogenes* and CIM gene knockdowns they confirmed the role for autophagy in controlling infection. All these processes were similar in CIMs and BMMs.

*M. tuberculosis*: Here, they infected BMMs and CIMs with auto-luminescent *M. tuberculosis* strains and looked at bacterial replication and report robust growth in CIMs with kinetics comparable to that of BMMs. To determine if CIMs could be used to explore the biology of tubercle bacteria, they assessed whether the specialized Esx secretion apparatus was important for replication in CIMS. For this, they use a *M. tuberculosis* strain lacking the core ATPase, required for this system, and report that the mutant bacteria were unable to replicate in BMMs or CIMs. Next, they explore the role of IFNγ and report similar outcomes between the two cell types. They also compared gene expression of BMMs and CIMs infected with wild type *M. tuberculosis* or a *ΔeccC* muant using the 700 gene panel described above. They find a strong correlation between genes expressed in CIMs and BMMs, thus confirming the utility of the CIM model for TB studies.

Key finding: The development of the Cas9+CIM system is a significant technological advance. It provides an invaluable tool for the study of macrophage biology and macrophage/pathogen interaction. Availability of this technology to the broader scientific community would help in the advancement of science.

Essential revisions:

1) One of the major concerns is choosing technical duplicates from one experiment for NanoString experiment. The authors should justify why they haven't performed biological replicates if they are not able to repeat the experiment in Figure 1C, Figure 3C, D.

---

## [Author Response]

Essential revisions:1) One of the major concerns is choosing technical duplicates from one experiment for NanoString experiment. The authors should justify why they haven't performed biological replicates if they are not able to repeat the experiment in Figure 1C, Figure 3C, D.

We have confirmed the results from the original transcriptional profiling experiment with a biological replicate, using the same NanoString platform as the readout. This new data demonstrates that we are able to replicate this result precisely, indicating that two cell populations are indeed quite similar in their transcriptional profiles. Representative data from the two experiments is displayed in Figure 1C, Figure 3C,D, and all data is available in the Figure 3—source data 1. We were also able to compare the transcriptional profile of Mtb-infected RAW 264.7 cells with BMMs and CIMs and have included these data in Figure 3—figure supplement 1.